# Tribological Properties of Carbon Fiber-Reinforced PEEK against 304 Stainless Steel with Reticulate Surface Texture

**DOI:** 10.3390/ma15248789

**Published:** 2022-12-09

**Authors:** Zhiyi Jin, Xifang Zhang, Zhibao Hou, Zhenqiang Yao, Hong Shen

**Affiliations:** 1School of Mechanical Engineering, Shanghai Jiao Tong University, Shanghai 200240, China; 2State Key Laboratory of Mechanical System and Vibration, Shanghai 200240, China

**Keywords:** PEEK composites, carbon fiber, friction coefficient, surface texture, tribofilm

## Abstract

With the aim of improving the durability and reliability of polyetheretherketone (PEEK) composites reinforced with carbon fiber (CF) as thrust bearings without lubricants, a reticulate surface texture was fabricated by plane honing on a stainless steel (SS) counterpart to promote its tribological properties. Pin-on-disk experiments were designed, with the results showing that the reticulate surface texture effectively reduces the friction coefficient from 0.40 to 0.20 compared with the polished SS surface, within the range of the *pv* value from 0.185 to 1.85 MPa∙m/s. The wear mechanism of the polished SS surface against CF-PEEK, proven with SEM and EDS observations as well as AE measurements, is revealed, falling into abrasive wear with SS particles embedded in the friction interface around the CF strips, causing three-body contact. The reduction in the friction coefficient of the textured SS disk against the CF-PEEK pin can be achieved due to diminution of the CF wear debris and SS particles, which are scraped off by the groove edges and trapped by the groove valleys, reducing the three-body abrasive wear, while the honed plateau is used as a flank surface like a cutting tool to scratch more soft PEEK particles as the transferred film, owing to adhesive wear. This investigation suggests that the SS disk with a honed surface structure can be used as the counterpart of CF-PEEK bearings with a low friction coefficient and wear rate under dry friction.

## 1. Introduction

Polyetheretherketone (PEEK) is popularly applied in the aerospace [1], medical [2,3] and industrial fields [4], owing to its extraordinary mechanical and chemical properties. To enhance its frictional and wear performances, PEEK is compounded with glass fiber (GF) or carbon fiber (CF) to prepare reinforcement materials [5,6]. Nevertheless, PEEK still suffers severe wear, which challenges the durability and reliability of PEEK components, as they are subjected to high temperature, heavy load and a lack of lubricant.

To improve the tribological performance of mechanical components, various techniques have been proposed to prepare textures on surfaces, such as laser surface texturing [7] and lithography–electrolytic technology [8]. Li et al. [9] investigated the friction coefficient of the laser-processed surface texture and found that the groove surface texture had the effect of reducing friction. Wu et al. [10] fabricated parallel groove surface textures on ASTM 1045 steel and observed that the interface friction temperature rise in textured specimens was significantly smaller. An experimental study by Menezes et al. [11] showed that the friction coefficient is highest when the surface texture is perpendicular to the direction of sliding. Conradi et al. [12] claimed that the texture with wear particle traps is effective under dry sliding conditions, such as grooves oriented perpendicular to or at 45° to the moving direction. Kang et al. [13] established the relationship between the adhesion work (contact angle) and the surface texture. The nanotextured surface has a small contact angle, indicating a high friction coefficient. Dong et al. [14] assumed that the friction reduction mechanism of PEEK against WC-Co under dry friction is that the debris that develops into a self-lubricious tribofilm counteracts the microcutting effect of the surface texture. Yang et al. [15] proposed that the collision frequency between friction counter faces is the main factor affecting the tribological properties of grooved surfaces. Cao et al. [16] indicated that the wear debris removed from the PEEK pin due to the microcutting effect of the dimple texture developed into tribofilms by repetitive friction cycles. Wang [17] designed and prepared circular dimples on polymer specimens, which revealed a stable friction coefficient and lower wear rates. Hammouti et al. [18] reported that surfaces with dimples exhibited an effective reduction in wear, although the surface pattern induced abrasive wear on the counterpart. Wang [8] fabricated a surface texture on stainless steel using lithography–electrolytic technology and concluded that the wear behavior was controlled by the elastic modulus of soft materials and the parameters of the surface pattern. These investigations clarified the friction reduction effects of surface texture, such as dimples and grooves fabricated by lasers or photolithography. However, the lithography process is complex and time-consuming, and the laser process might introduce a thermal effect, which limits its wide application in industry.

Due to the high efficiency and low cost of the honing process, it is widely used to reduce friction loss [19] by fabricating a reticulated texture on the inner wall of the cylinder hole. Hu et al. [20] experimentally studied the surface morphology and tribology property evolution of cylinder liners under mixed lubrication, and the results showed that a slide-honed surface is beneficial to reduce wear. Woś et al. [21] demonstrated significant differences in engine performance due to different initial cylinder liner roughness parameters. The previous studies are mainly focused on tribology behavior of honed surface structure in liquid lubrication. For the bearings used in equipment, such as pumps subjected to frequent start and stop, the bearings will suffer from boundary lubrication, and even dry friction, where the friction behavior between CF-PEEK bearings and the counterparts plays an important role in influencing the reliability of the equipment. The studies on tribological properties and the formation of tribofilm under dry friction are rarely reported.

In the present study, two reticulated textures were prepared by plane honing on the surface of SS disks, and the tribological characteristics were examined using a pin-on-disk test machine to enhance the tribological properties of 304 SS against carbon fiber-reinforced PEEK (CF-PEEK) under dry friction, as well as polished SS disks and ground SS disks as controls. The friction coefficient and acoustic emission (AE) signal were recorded throughout the experiments. The surface roughness parameters of the SS disks were analyzed to reveal the influence on the friction coefficient and the AE signals. Scanning electron microscopy (SEM) was applied to observe the worn surface of the SS disks and CF-PEEK pins. Energy-dispersive spectroscopy (EDS) was used to detect the formation of the PEEK transfer film on the SS disk and the element component of wear debris embedded on the CF-PEEK pin, in order to investigate the wear mechanisms of different surface patterns.

## 2. Experiments

### 2.1. Sample Preparation

The frictional behavior was experimentally evaluated using a pin-on-disk test machine, where CF-PEEK was used as the pin and 304 SS was selected as the disk. CF-PEEK pins with a diameter of 6 mm were made from commercial rods provided by Ensinger (Nufringen, Germany). The main material parameters of CF-PEEK are listed in Table 1. To ensure similar surface conditions, the end face of each pin was precisely turned by INDEX G200 (Esslingen, Germany) with a surface roughness around 0.40 μm. Before the experiments, all disks and pins were ultrasonically cleaned with alcohol for 30 min and dried.

Four kinds of surface conditions were prepared in the experiment, including two kinds of textured surfaces, as well as a polished surface and ground surface as controls. The polished surface was fabricated using a polishing machine (MECATECH 250 SPI, PRESI, Eybens, France) to ensure an average surface roughness at around 0.02 μm. The ground surface was manufactured using a high-speed grinding machine (MT408, BLOHM, Hamburg, Germany) to guarantee an average surface roughness of approximately 0.80 μm. The polishing machine was also used to prepare the reticulated texture on the surface of the specimens, as shown in Figure 1. The diameter of the polishing platen $d_{1}$ is 250 mm. The envelope of the abrasive paper trajectory in the workpiece coordinate system is a ring. To enable the surface pattern to cover the whole surface of the workpiece, the eccentricity *e* and the position of sandpaper *r* should satisfy the following equations:(1)$$\left|r-e\right|\leq d_{2}-\frac{1}{2}d_{3}$$
(2)$$\left|r+e\right|\geq d_{2}+\frac{1}{2}d_{3}$$
where $d_{2}$ is the diameter of the workpiece and $d_{3}$ is the center distance between the workpiece and the holder.

By changing the rotational speed ratio of the specimen holder *n_w_* to the polishing platen *n_p_*, different densities of the surface textures can be obtained. In this paper, two kinds of surface textures were prepared, and the working parameters are listed in Table 2.

The morphologies of the four kinds of surface patterns were observed by a metallographic microscope (AxioCam ICc 5, ZEISS, Oberkochen, Germany), and the surface profiles were measured by a stylus measuring device (SJ-210, MITUTOYO, Kanagawa, Japan).

### 2.2. Tribology Test Method

The test rig (MFT-5000, RTEC, San Jose, CA, USA) was applied to examine the tribological behaviors of the specimens under different *pv* values, as shown in Figure 2. According to ASTM G99-95a, the processing parameters of the pin-on-disk test are listed in Table 3. The specific pressure *p* was adjusted from (0.707 to 2.83) MPa by changing the applied load from (20 to 80) N, while the velocity *v* was regulated from (0.26 to 0.65) m/s by selecting a different friction track to change the center distance between the pin and disc, as illustrated in Figure 3. Therefore, the total *pv* is controlled from (0.185 to 1.85) MPa∙m/s. To obtain the dynamic signal during friction, an AE sensor (GM150, Qingcheng, China) was installed on the back of the CF-PEEK pin to obtain the AE signals, as shown in Figure 3. During the test, the pin and the AE sensor are tightly fitted by locking the nut. To ensure the reliability of the experiments, at least three tests of each group were repeated.

### 2.3. Surface Morphology and Transfer Film of Stainless Steel

The morphologies of the worn SS disks and CF-PEEK pins were examined through SEM (VEGA 3, TESCAN, Brno, Czech Republic). EDS (INCA X-Act, Oxford, UK) was performed to detect the element composition on the surface of the SS disks and the CF-PEEK pins.

## 3. Results and Discussion

### 3.1. Surface Characteristics

The original workpiece surface topographies of polished, textured and ground are illustrated in Figure 4, and the analysis of the surface profiles is shown in Figure 5 correspondingly. It can be seen from Figure 5 that the first column (a, e, i, m) displays the surface profile curves; the second column (b, f, j, n) shows the BACs (bearing area curves, also called Abbott–Firestone curves), which describe the increase in the material portion of the surface with increasing depth of the roughness profile; the third column (c, g, k, o) exhibits the ADCs (amplitude distribution curves), which express the distribution density of material in the depth direction; and the fourth column (d, h, l, p) represents the Fourier transform results of the surface profile curves. The BAC is equivalent to the cumulative probability density function, while the ADC corresponds to probability density function in statistics. The Y-axis in BACs and ADCs represents the height of peaks and troughs, in which the highest peak is 0%, and the lowest trough is 100%. It can be concluded from the first column and the fourth column that the profile curve of the polished surface is the densest, and the proportion of high-frequency components is significantly higher than that of the other three functional surfaces. The polished surface and ground surface had an approximately symmetrical distribution of peaks and valleys (polished surface: −0.06 to 0.06 μm; ground surface: −3.0 to 3.0 μm), as shown in Figure 5b,c,n,o, compared with the peaks, and the valleys of the two textured surfaces are distributed asymmetrically (texture-1: −1.5~0.5 μm; texture-2: −2.5~0.5 μm), as shown in Figure 5f,g,j,k.

The average values of surface roughness *R*_a_ of the four kinds of surfaces are significantly distinct. The polished surface had the lowest *R*_a_ at 0.02 μm, while the *R*_a_ of the ground surface is the highest at about 0.8 μm. The *R*_a_ values of the two textured surfaces are 0.14 μm and 0.26 μm, respectively. According to ISO 13565-2, the surface characterizations of the Abbott–Firestone curve are listed in Table 4, where *R*_pk_ is the reduced peak height, *R*_k_ is the core roughness, *R*_vk_ is the reduced valley depth, *M*_r1_ is the material ratio 1 at the peak zone, and *M*_r2_ is the material ratio 2 at the valley zone.

### 3.2. Frictional Behaviors

The typical evolution of the friction coefficient under different *pv* values from 0.185 to 1.85 MPa∙m/s are illustrated in Figure 6, and the mean value and standard deviation of the friction coefficient are shown in Figure 7. It can be indicated from Figure 7 that the polished surface has the highest friction coefficient, at around 0.40. The low surface roughness of the polished surface (*R*_a_ = 0.02 μm) leads to a high adhesive component of the tribological coupling [22]. This result is also attributed to the distribution of peaks and valleys on the surface for diverse structured surfaces. Table 4 shows that the polished surface has a small *R*_pk_ value at 0.015 μm; however, the other three surfaces have higher *R*_pk_ values at 0.16 μm, 0.13 μm and 1.0 μm. Moreover, the polished surface also has the highest density of the surface profile curve (see Figure 5a,d), which explains why the polished surface has smaller surface asperities. This also indicates that the microconvex with extremely small size is distributed on the polished surface. Therefore, during the sliding process, the small hard SS particles fall off by the physical interlocking between the polished surface asperities [23] and embed into the contact surface between CF-PEEK blends and the SS disk, resulting in three-body wear. In addition, the polished surface has a small *R*_vk_ value at −0.1 μm and a large *M*_r2_ value at 90.5%, while the other three surfaces have higher *R*_vk_ values at 0.44 μm, 0.92 μm and 1.35 μm and lower *M*_r2_ values at 83.4%, 79.8% and 87.8%, respectively, as listed in Table 4. It is inferred that the chip storage capacity of the polished surface is lower than that of the other three surfaces. There is some wear debris sliding or rolling randomly between the polished SS disk and CF-PEEK pin, leading to the friction coefficient fluctuating significantly with increasing processing time, as shown in Figure 6a. The mean friction coefficient value of the polished SS disk fluctuates between 0.38 and 0.42 as the *pv* value increases from 0.185 to 1.85 MPa∙m/s, which means that the friction coefficients of the smooth, hard SS surface against the rough, soft CF-PEKK surface remain almost unchanged with the load.

Figure 6b,c show the friction evolution of the two kinds of textured SS surfaces. The friction coefficients of the two kinds of textured surfaces are significantly smaller (at approximately 0.20) and more stable than those of the polished surface with increasing *pv* value. It is suggested that the grooves on the textured surface can inhibit the physical interlocking effect between the asperities on the polished surface and reduce adhesive wear. According to the Abbott–Firestone curves in Figure 5f,j, two textured SS disks have an asymmetric distribution of peaks and valleys on surfaces, as well as lower *M*_r2_ values of 83.4% and 79.8%, which implies that the textured surface with microgrooves has better chip storage capacity. Some CF-PEEK wear debris, especially CF debris, are captured by grooves reducing the three-body abrasive wear; therefore, PEEK particles are transferred to a tribofilm on the textured SS surface and serve as solid lubricants, resulting in a smaller friction coefficient on the textured surface, as illustrated in Figure 8. Furthermore, under the condition of a large *pv* value of 1.11 to 1.85 MPa∙m/s, the friction coefficient of tuexured_2 is slightly smaller than that of textured_1, owing to the lower *M*_r2_ of the textured surface obtained with the higher texture densities and larger valley depth. It is considered that the wear debris is more easily captured by groove textures on the surface with a lower value of *M*_r2_ during movement.

The friction evolution of the ground surface is shown in Figure 6d. The friction coefficient of the ground surface is much lower than that of the polished surface, as illustrated in Figure 7. It is believed that grinding scratches on the ground surface have the capacity to store wear debris, thus reducing abrasive wear impact [24]. The friction coefficient of the ground surface is slightly higher than that of the textured surface, especially under low *pv* value. The *R*_a_ values of the two textured surfaces are 0.14 μm and 0.26 μm, which are smaller than that of the ground surface at 0.80 μm. The mechanical effects caused by the scratching of rough surfaces are dominant under low loads. When the applied load is high, due to the main wear mechanism transferring from scratch to abrasive wear, the advantage of the chip storage capacity of the ground surface is magnified. The friction coefficient of the ground surface decreases gradually from 0.26 to 0.18 as the *pv* value varies from 0.185 to 1.85 MPa∙m/s. It can be inferred from Figure 9 that the peaks on the ground SS surface are smoothed by CF-PEEK with a larger *pv* value during the friction test, and the distribution of peaks and valleys on the ground surface become asymmetric, similar to the two textured surfaces. In addition, the thermal effects accompanied by heavy load and high speed play an important role in the rheological properties of polymers [25], resulting in a downward trend of friction coefficients with the sliding speed.

### 3.3. AE Signals

The surface topography has a great influence on the AE root mean square (RMS) [26], which was collected to further investigate the tribological properties during the experiment, as shown in Figure 10. The AE RMS can detect the fracture of material and the movement of wear debris, which emit a high-level acoustic sound [27]. It can be obtained from Figure 11 that the RMS value increased significantly with increasing *pv* value for polished, textured and ground surfaces, indicating the increase in asperity elastic–plastic deformation, materials exfoliation and wear particle motion. In addition, the amplitude of the AE signal fluctuates greatly for the polished surface during the frictional experiment, which is almost synchronous with the fluctuation of the friction coefficient referred to Figure 6a and Figure 10a. Wear debris accumulation on the friction interface will deteriorate the friction-induced vibration and noise behavior of the friction system [28]. It is believed that the wear debris falls off from the polished SS disk and CF-PEEK via adhesive wear, and the movement of wear particles between the friction interface produces severe elastic-plastic deformation of materials, inducing high RMS values.

The RMS signal of the two kinds of textured surfaces increased during the running-in period and then decreased to saturation, as shown in Figure 10b,c. It is because CF-PEEK wear debris continuously fell off from the substrate and generated a large acoustic emission signal under the microcutting effect of the textured surface at the beginning of the experiments. Subsequently, part of the abrasive particles is embedded in the groove texture with the running-in period, which reduces the microcutting effects, and others develop into a transfer film, serving as solid lubricants. Because the *R*_a_ of textured_1 is smaller than that of textured_2, the maximum amplitude of the AE signal of textured_1 is smaller than that of textured_2, as exhibited in Figure 11. Consequently, the wear debris peeled by the microcutting effect of the textured surface is relatively small, resulting in a lower AE RMS value.

For the ground surface, the AE signal remains relatively stable when the *pv* value is small, while the amplitude and fluctuation of the signal increase when the *pv* value increases, as shown in Figure 10d. The RMS value fluctuates violently during the running-in period, which is due to the peaks on the ground SS surface being gradually smoothed by the CF-PEEK pin, making the surface characteristics close to the two texture surfaces, as shown in Figure 9c. It is supposed that large amounts of mechanical scratches exist between the CF-PEEK pin against the relatively rough ground SS surface, resulting in a significant increase in abrasive wear with a relatively high RMS value, especially under heavy loads.

### 3.4. Surface Morphology and Transfer Film

The surface morphologies of the worn SS disks and CF-PEEK pins were examined using SEM and the element contents on the surfaces were detected by EDS, as shown in Figure 12 and Figure 13. Several lines parallel to the friction direction are marked with white arrows in Figure 12a, indicating that there are scratches caused by abrasive wear on the polished SS disk. The EDS map of the polished surface specimen shows that there is almost no carbon element on the surface, which represents the main element of CF-PEEEK, while some iron element with a mass fraction of approximately 4.0% is found on the surface of the CF-PEEK pin, as shown in Figure 12b and Figure 13b. It is verified that the hard SS asperities are scraped off by the CF-PEEK pin and embedded in the soft polymer. It can be indicated that there is almost no transfer film on the polished SS surface; however, the three-body abrasive wear caused by SS debris has a significant effect on the high friction coefficient and AE RMS value of the polished surface.

There are almost no scratches parallel to the friction direction on the textured SS surface; nevertheless, the content of carbon element on the textured SS (with a mass fraction of about 16%) is much higher than that on the polished surface, especially near the grooves, as illustrated in Figure 12d,f. There are amounts of pores on the surface of the worn CF-PEEK pin, as shown in Figure 13d,g, as well as almost no iron element on the surface of CF-PEEK pins, as shown in Figure 13e,f,h,i. It is inferred that CFs may be pulled out from the substrate under the effect of the SS texture microcutting effect. However, the CF wear debris will be captured by the groove, reducing the abrasive wear, and other PEEK particles will further transfer to a tribofilm on the surface serving as solid lubricants, thus leading to adhesive wear.

Large amounts of debris can be seen on the ground SS surface, resulting in the highest content of carbon element on the surface, as shown in Figure 12g,h. There are some abrasive debris accumulated on the ground surface, which reduced the mechanical scratch to some extent. In addition, there is an obvious mechanical scratch with almost no iron element on the surface of the CF-PEEK pin, as shown in Figure 13j–l. It is considered that the main wear mechanism is the mechanical scratching of the rough SS surface on CF-PEEK. Therefore, the relatively rough ground surface leads to large numbers of mechanical scratches and wear rates of the CF-PEEK counterpart, resulting in a higher AE RMS value and larger fluctuation of the friction coefficient than the textured surfaces.

### 3.5. Influence of Morphology on Tribology Behavior

The sketches of the wear mechanism when CF-PEEK pins slide against textured SS disks, polished and ground disks are proposed and illustrated in Figure 14. As shown in Figure 14a, when the CF-PEEK pin slides against the polished SS disk which has the lowest surface roughness as shown in Figure 5a,d, the worn debris of PEEK, CF and SS hard particles exist between the friction interfaces. When some SS hard particles are embedded into the PEEK matrix, others remain between the interface due to the support of CF strips enclosing the hard particles, resulting in high friction force in the sliding direction (as shown in Figure 6a and Figure 7) and scratch scars on SS disks (as shown in Figure 12a). It is proven by the EDS maps (see Figure 13a) that iron element exists on the surface of CF-PEEK pins.

When the CF-PEEK pins slide against the SS disks with the reticulated grooves on polished substrate, the transferred PEEK film can be observed on SS disks, which was proven with EDS maps (as shown in Figure 12d,f) without iron element being found on the CF-PEEK pins (as shown in Figure 13e,h). The interaction mechanism between CF-PEEK pins and SS disks is proposed and modeled in Figure 14b, where the honed plateau can be regarded as the flank surface of cutting tools to scratch more soft PEEK particles as the transferred film due to adhesive wear, while the groove wall in front of the plateau serves as the rake face of cutting tools to scratch off worn debris stored in the groove valley, including SS hard particles, CF strips and PEEK particles (as shown in Figure 12d,f), resulting in a lower friction coefficient (as shown in Figure 6b,c) without scratch scars on SS disks (as shown in Figure 12c,e).

As the CF-PEEK pins slide against the ground SS disks, the friction coefficient is close to the honed SS disks, while the CF-PEEK pins suffer severe wear as compared to the polished disk surface. EDS maps exhibit the existence of carbon element in the grinding scratch on the SS disk surface as shown in Figure 12h, indicating the storage of CF-PEEK debris. The interaction process between CF-PEEK pins and SS disks is proposed and modeled in Figure 14c, where a serrate cross section with a larger valley than polished surface morphology can be utilized to collect worn debris, avoiding three-body contacting of hard particles, resulting in a lower friction coefficient. The large groove ratio at the disk surface accounts for the high wear rate of CF-PEEK pins against ground SS disks, which can be proven in Figure 12. The peaks of serrate texture will be smoothed by CFs, as illustrated in Figure 9, resulting in a decreasing friction coefficient close to the honed disks, as shown in Figure 6d.

## 4. Conclusions

In this work, the tribological properties of CF-PEEK against SS with four kinds of surface patterns were investigated, with AE signals being collected throughout the friction process. The worn surface morphologies of SS disks and CF-PEEK pins were observed by SEM, and the element distributions on the surfaces were analyzed by EDS to reveal the wear mechanism and clarify the main reason for friction reduction by using the reticulated surface texture. The main conclusions are summarized as follows:(1)The reticular texture on the SS surface fabricated by the plane honing process can reduce the friction coefficient against CF-PEEK significantly under dry friction conditions from approximately 0.40 to about 0.20, compared with the polished SS surface. It is revealed and proven with EDS that the abrasive wear mechanism dominates the three-body contact of the polished SS disk against the CF-PEEK pin, with hard SS particles between the interface accumulating near the CF strips on the pin.(2)The AE RMS value increases significantly with increasing *pv* value for polished, textured and ground surfaces, indicating an increase in elastic–plastic deformation and wear rates. The AE RMS value of the polished SS disk is the highest, while that of the textured SS disk is the lowest, indicating that the smooth contact surface suffers sever elastic–plastic deformation and abrasive wear, which verifies the antiwear property of the textured SS surface.(3)The ground SS surface shows a low friction coefficient under a large *pv* value, due to the peaks on the ground SS surface being smoothed by CF-PEEK during the friction test with a high wear rate of the CF-PEEK counterparts. The distribution of peaks and valleys on the ground SS disk surface becomes asymmetric, similar to the two textured surfaces, resulting in a reduction in the friction coefficient.(4)The reduction of the friction coefficient with CF-PEEK against textured SS disks can be attributed to diminution of the CF wear debris and SS particles, which are scraped off by groove edges and trapped by groove valleys, reducing the three-body abrasive wear, and promotion of PEEK particles transferred to a tribofilm on the SS surface due to the flank surface effect of the grooved plateau under adhesive wear. The results show that the plane honing technique has the potential to be applied to the runner fabrication of thrust bearings facing the possible risk of dry friction.

## Figures and Tables

**Figure 1 materials-15-08789-f001:**
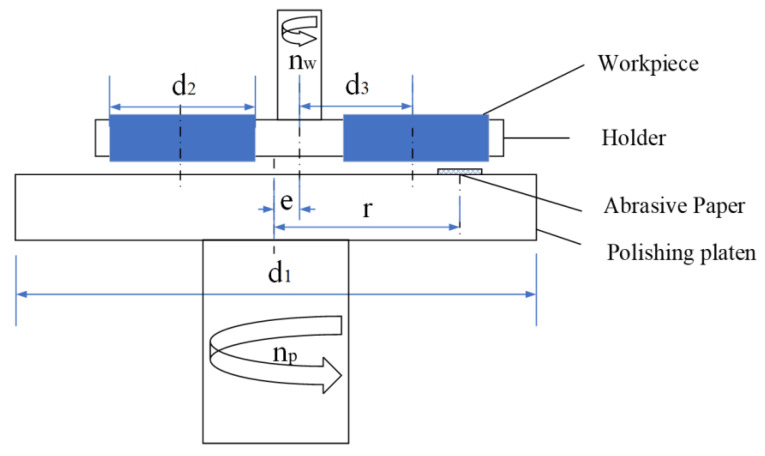
Schematic diagram of the preparation of the reticulated texture on the specimens.

**Figure 2 materials-15-08789-f002:**
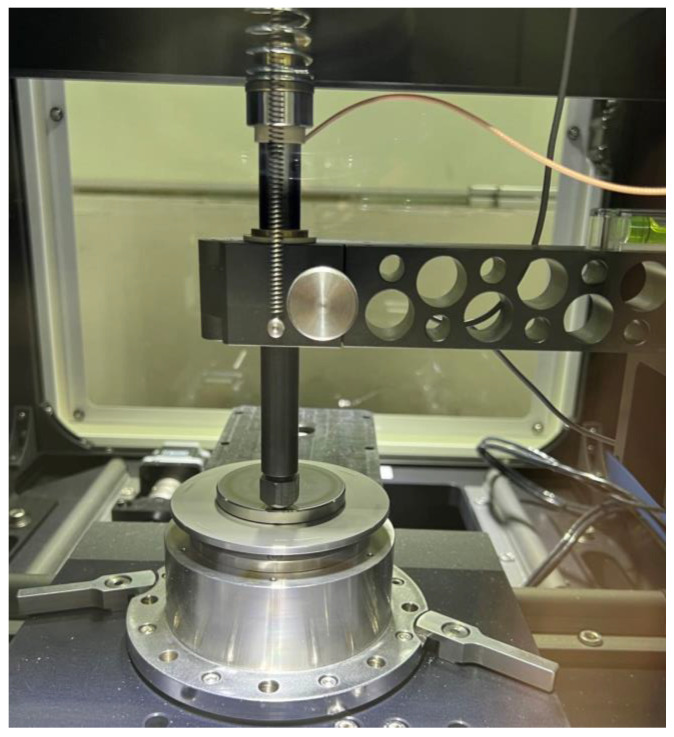
Tribology performance test on the MFT-5000 test rig.

**Figure 3 materials-15-08789-f003:**
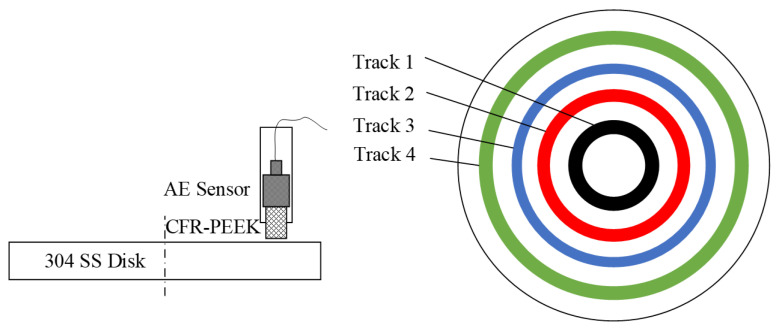
Schematic diagram of the friction test and AE sensor.

**Figure 4 materials-15-08789-f004:**
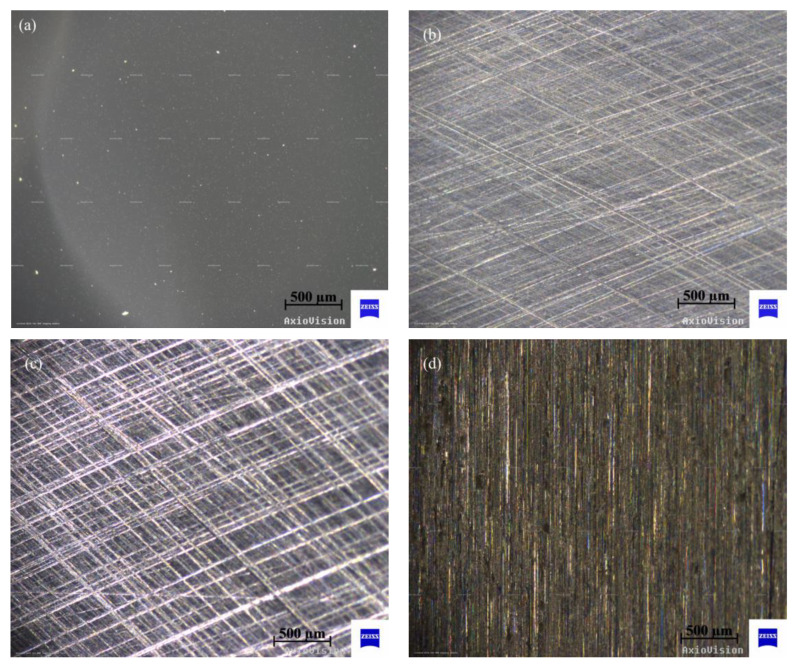
Original morphologies of four kinds of surfaces: (**a**) polished surface; (**b**) textured surface −1; (**c**) textured surface −2; (**d**) ground surface.

**Figure 5 materials-15-08789-f005:**
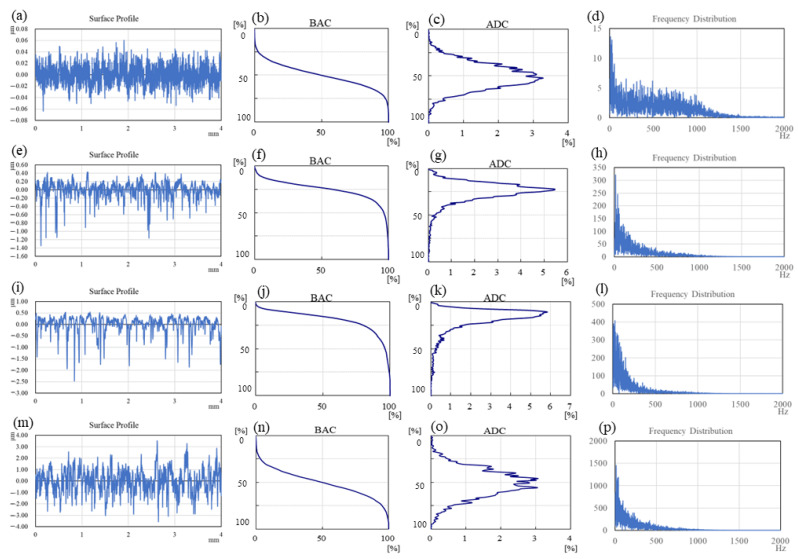
Analysis of the initial surface profiles: (**a**–**d**) polished surface; (**e**–**h**) textured surface −1; (**i**–**l**) textured surface −2; (**m**–**p**) ground surface.

**Figure 6 materials-15-08789-f006:**
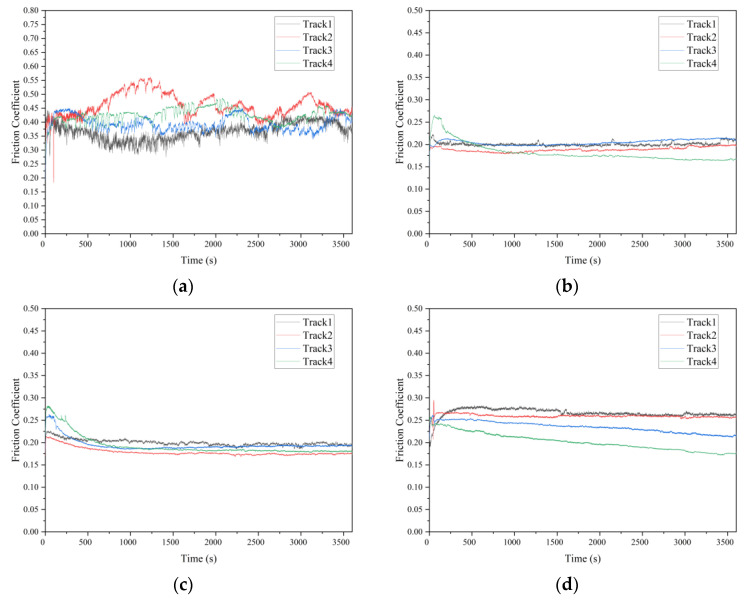
Friction coefficient evolution versus sliding time under different *pv* values: (**a**) polished surface; (**b**) textured surface −1; (**c**) textured surface −2; (**d**) ground surface.

**Figure 7 materials-15-08789-f007:**
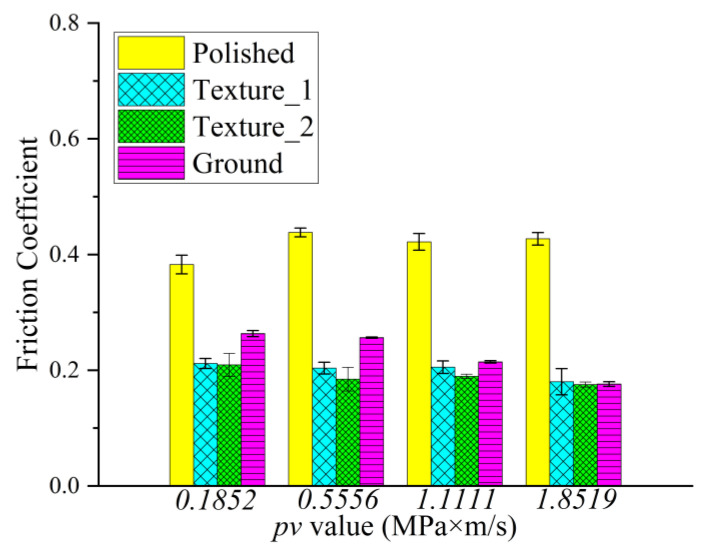
Mean and standard deviation of the friction coefficient of four kinds of surface patterns in three repeated experiments under different *pv* values.

**Figure 8 materials-15-08789-f008:**
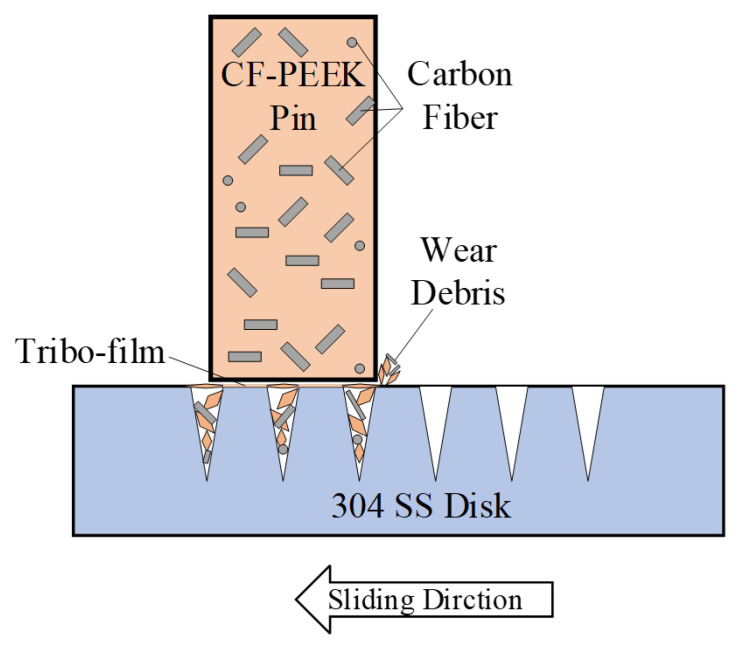
Schematic diagram of the friction reduction mechanism by surface texture.

**Figure 9 materials-15-08789-f009:**
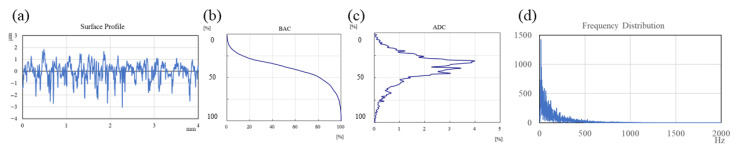
Analysis of the worn ground SS disk surface: (**a**) the surface profile, (**b**) the material ratio curve, (**c**) the amplitude distribution curve, (**d**) the frequency distribution curve.

**Figure 10 materials-15-08789-f010:**
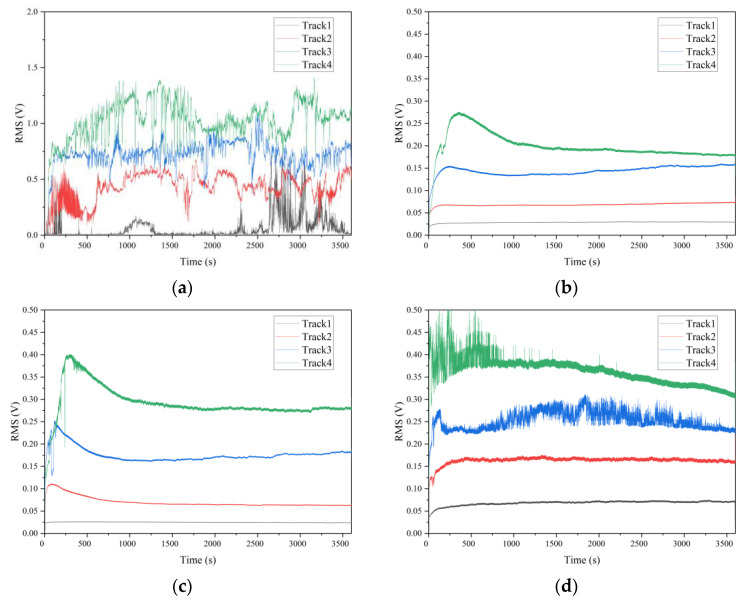
AE RMS signal evolution versus sliding time under different *pv* values: (**a**) polished surface; (**b**) textured surface −1; (**c**) textured surface −2; (**d**) ground surface.

**Figure 11 materials-15-08789-f011:**
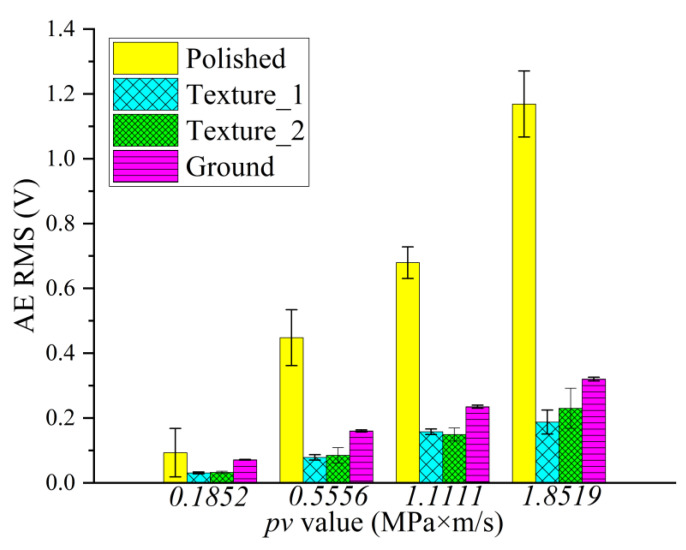
Mean and standard deviation of AE RMS values of four kinds of surface patterns in three repeated experiments under different *pv* values.

**Figure 12 materials-15-08789-f012:**
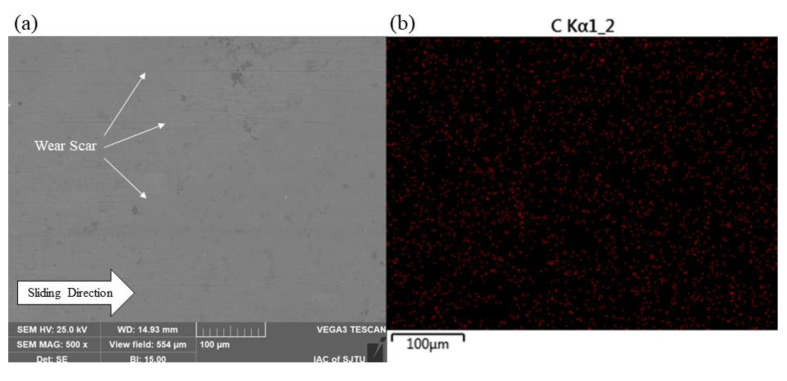
SEM micrographs of worn SS disks and carbon element distribution on surface: (**a**,**b**) polished surface; (**c**,**d**) textured surface −1; (**e**,**f**) textured −2; (**g**,**h**) ground surface.

**Figure 13 materials-15-08789-f013:**
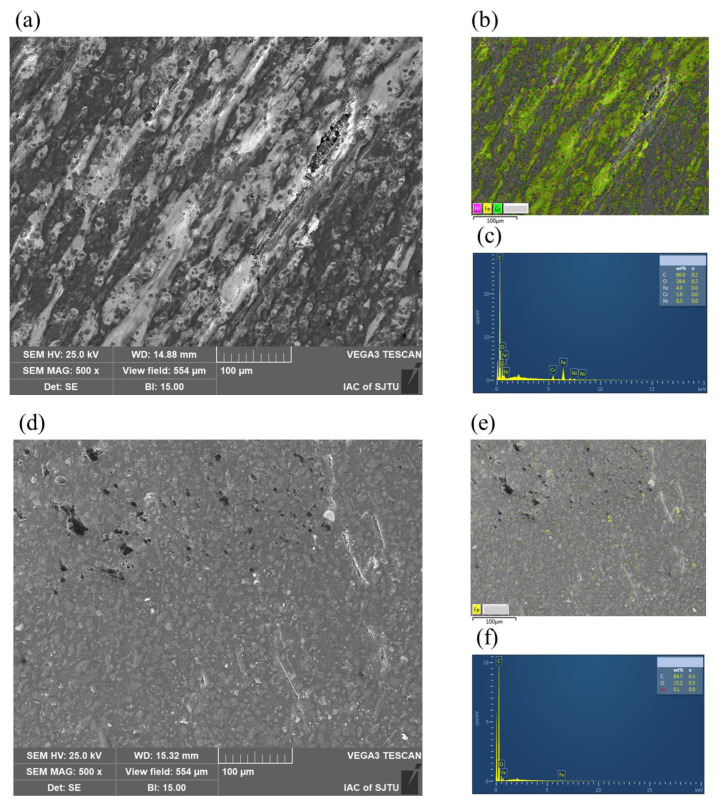
SEM micrographs of worn CF-PEEK pins and EDS results of CF-PEEK surface against SS disks with: (**a**–**c**) polished surface; (**d**–**f**) textured surface −1; (**g**–**i**) textured surface−2; (**j**–**l**) ground surface.

**Figure 14 materials-15-08789-f014:**
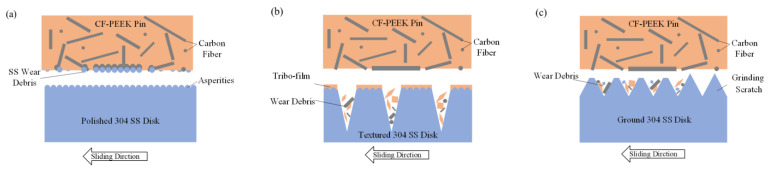
Sketch of wear mechanism when CF-PEEK pins slide against (**a**) polished SS disks; (**b**) textured SS disks; (**c**) ground SS disks.

**Table 1 materials-15-08789-t001:** Material properties of CF-PEEK.

Properties	Modulus of Elasticity (MPa)	Ball Indentation Hardness (MPa)	Density (g/cm^3^)	Melting Point (°C)	Glass Transition Temperature (°C)	Thermal Conductivity (W/m × K)
CF-PEEK	6000	298	1.38	341	147	0.66

**Table 2 materials-15-08789-t002:** Operating conditions of surface texture preparation.

Parameters	*n_w_*/rpm	*n_p_*/rpm	*e*/mm	*r*/mm	*d*_1_/mm	*d*_2_/mm	*d*_3_/mm
Texture_1	20	51	30	50	250	50	60
Texture_2	20	201	30	50	250	50	60

**Table 3 materials-15-08789-t003:** Pin-on-disk test parameters (dry sliding).

Track Number	Unit	1	2	3	4
Diameter of the tracks	mm	20	30	40	50
Sliding velocity	m/s	0.2618	0.3927	0.5236	0.6545
Sliding distance	m	942	1414	1885	2356
Applied load	N	20.00	40.00	60.00	80.00
Specific pressure	MPa	0.7074	1.4147	2.1221	2.8294
*pv* value	MPa∙m/s	0.1852	0.5556	1.1111	1.8519

**Table 4 materials-15-08789-t004:** Parameters of the Abbott–Firestone curve.

Parameters	Polished	Textured_1	Textured_2	Ground
*R*_pk_/μm	0.015	0.161	0.130	1.002
*R*_k_*/*μm	0.04	0.345	0.566	2.679
*R*_vk_*/*μm	0.016	0.438	0.916	1.347
*M*_r1_/%	9.013	7.913	2.938	8.544
*M*_r2_/%	90.475	83.409	79.800	87.813

## Data Availability

Data sharing not applicable.

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
