# Peer review of "Tribological Properties of Carbon Fiber-Reinforced PEEK against 304 Stainless Steel with Reticulate Surface Texture"

_materials, 2022, doi:10.3390/ma15248789_

Round 1

Reviewer 1 Report

Dear Authors, after reading your manuscript, I think that your manuscript could be reconsidered for publication should you be prepared to incorporate major revisions.

What was the unique contribution of this work in comparison to other publications in your field?

What is the research gap did you find from the previous researchers in your field? Please note that the paper may not be considered further without a clear research gap and novelty of the study. Please underscore the scientific value-added to your paper in your abstract should clearly state the essence of the problem you are addressing, what you did and what you found and recommend.

The results are merely described and are limited to comparing the experimental observation. The authors are encouraged to include more detailed discussion and critically discuss the observations from this investigation with existing literature.

Please make sure your conclusions section underscores the scientific value-added of your paper, and/or the applicability of your findings/results. Highlight the novelty of your study.

Author Response

Dear reviewer,

  Thank you for your comments on our manuscript entitled " Tribological Properties of Carbon Fiber Reinforced PEEK against 304 Stainless Steel with Reticulate Surface Texture " (ID: materials-2074647). Those comments are very helpful for revising and improving our paper, as well as the important guiding significance to further research. We have studied the comments carefully and made corrections which we hope to meet with approval. The main corrections are marked in green color in the manuscript and the responds to the reviewers’ comments are as follows (the replies are highlighted in blue). The number of the figures, tables and references in the letter are the same as the manuscript revised.

Reviewer 2 Report

in this article "Tribological Properties of Carbon Fiber Reinforced PEEK 2

against 304 Stainless Steel with Reticulate Surface Texture" authors were able to fabricate surface texture on stainless steel in order to improve the durability and reliability of polyetheretherketone. The authors were able to reduce the friction coefficient significantly, from approximately 0.40 to about 0.20. Also, the  CF-PEEK during the friction was tested and the result has been presented. The article is well written and the results were sufficient however I have a few questions :

1- the direction of the Np in figure 1, what about if we move it to the opposite direction, Are the results going to be affected? can you do it and record the results .

2- Parameters of the Abbott–Firestone curve. what if we increased the published ? is the results can be improved. 

3- can you increase the velocity to record the results?

thank you 

Author Response

(The authors gave the same response as above.)

Reviewer 3 Report

The article deals with an interesting and topical problem of carbon fibre and stainless steel materials in terms of tribology. For the research itself, the authors chose the pin-on-disk method as described in ASTM G99-95a standard. However, this standard is not cited anywhere in the text, neither is it listed among the references.

The strength of the article is a clearly presented research and an overview of who and what has been achieved in the issue under discussion. A schematic of the experimental setup is presented in the paper to give a clearer idea of the experimental setup. In my opinion, the strength of the paper is the application of the chosen methods in the paper and hence the results obtained. These results then provide a variety of interesting information from the experiments performed and their interpretations. The thesis is clearly written.

On the other hand, several formal errors or missing citations to the standard mentioned above are a weakness. For the ease of understanding, I list below the shortcomings that should be corrected or supplemented:

Line 77 - you state that the surface has been precisely machined, it is advisable to add by what method;

Line 78 - you state that all samples were cleaned in alcohol before testing, does this include the pin? If so, it is appropriate to add;

Line 101 - the marking of the microscope AxioCam ICc 5, ZWISS, Germany is wrong, the correct one is: AxioCam ICc 5, ZEISS, Germany, to be corrected;

Line 111-114 - ranges of values are usually indicated in parentheses so that it is clear that the unit refers to both values, e.g. 20 to 80 N, better is (20 to 80) N;

Line 114 - the unit MPa×m/s is inappropriately written, a better form is MPa∙m⁄s, this also applies to Table 3, also lines 161, 182, 197 and 209

Figure 5. Although the paper contains an interestingly designed experiment and results. However, it would be useful to explain in the introduction or in the description of the experiment what Abott-Firestone curves are. What do these curves imply, what do they express, and what scale are they given, this is missing from some of the graphs! This explanation should also be made with reference to amplitude distribution curves. Describe how the representation of peaks and troughs is expressed in ADCs, e.g. compare with the Gaussian distribution. It is useful to briefly address this explanation in the paper so that the not fully knowledgeable reader is clear what the graphs express and what information they give.

Figure 13 a) The first third of the figure shows an obvious scanning error during electron microscope documentation (horizontal line). This error commonly happens by varying the voltage during scanning. Usually the scanning of the image is done again if the image has to be published. Images without these errors are usually published in journals. If you have another better image, it is advisable to replace it or re-document it.

In conclusion, I positively evaluate the overall summary in four points. However, it is worth mentioning the practical application of the results. Specifically, where the results can be applied in industry, practice, production or even further research and development.

Author Response

(The authors gave the same response as above.)

Reviewer 4 Report

The manuscript is very interesting. There is only a minor comment: Caption in Figure 13 should also mention that the figure contains the EDS results.   

Author Response

Dear reviewer,

  Thank you for your comments on our manuscript entitled " Tribological Properties of Carbon Fiber Reinforced PEEK against 304 Stainless Steel with Reticulate Surface Texture " (ID: materials-2074647). Those comments are very helpful for revising and improving our paper, as well as the important guiding significance to further research. We have studied the comments carefully and made corrections which we hope to meet with approval. The main corrections are in green color in the manuscript and the responds to the reviewers’ comments are as follows (the replies are highlighted in blue). The number of the figures, tables and references in the letter are the same as the manuscript revised.

Round 2

Reviewer 1 Report

Dear Authors, thanks for considering and discussing all my mentioned points. I see a significant improvement in your manuscript and I recommended to accept the paper. Thanks for sharing your research.

Best regards.

Author Response

Dear reviewer,
  Thank you again for your comments on our manuscript helping us to improve our paper. 
Best regards.